# A pattern of platelet indices as a potential marker for prediction of pre-eclampsia among pregnant women attending a Tertiary Hospital, Ethiopia: A case-control study

**Solomon Gebre Bawore[1]***, **Wondimagegn Adissu**[2], **Berhanu Niguse[3], Yilma Markos Larebo[4], Nigussie Abebe Ermolo[5], Lealem Gedefaw[2]**

**1** Department of Medical Laboratory Sciences, College of Medicine and Health Sciences, Wachemo University, Hosaena, Ethiopia, **2** School of Medical Laboratory Sciences, Institute of Health, Jimma University, Jimma, Ethiopia, **3** Wachemo University Nigist Eleni Mohammed Memorial Comprehensive and Specialized Hospital, Hadiya Zone, Southern Nation Nationality People Region, Hosaena, Ethiopia, **4** Department of Epidemiology, College of Medicine and Health Sciences, Wachemo University, Hosaena, Ethiopia, **5** Nigussie Abebe Ermolo, Department of Health Service Management, College of Medicine and Health Sciences, Wachemo University, Hosaena, Ethiopia

* solomongebre16@gmail.com, solomongebre@wcu.edu.et

**Data Availability Statement:** All relevant data are within the paper.

## Abstract

### Introduction

Preeclampsia is the most serious health risk during pregnancy for both the mother and the fetus. Even though platelet parameters are among the proposed biomarkers for the prediction of preeclampsia, the use of its indices in the diagnosis of preeclampsia is not increasing in Ethiopia. There is little information on platelet patterns in preeclampsia and normal pregnancy. The purpose of this study was to determine the pattern of platelet indices in women with preeclampsia in our study setting.

### Methods

A case-control study was conducted among 180 pregnant women who attended anti-natal follow-ups from January 1 to April 3, 2019. An Ethylene Diamine Tetra Acetic Acid anti-coagulated venous blood was collected and analyzed using a hematology analyzer (MIND-RAY®-BC-300Plus, Shenzhen China). The SPSS software version 26 was used to run the Mann Whitney U test, Kruskal-Wallis H test, and Kolmogorov-Smirnov normality test, Post-hock test augmented with Benforeni, receiver operating characteristics curve, and Spear Man rank-order correlation. A P-value of <0.05 was considered statistically significant.

### Results

A total of 180 pregnant women were included in the study. Platelet count and platelet crit levels tend to decrease as pre-eclampsia becomes more severe. In contrast, the mean platelet volume and platelet distribution widths were significantly increased with the severity of pre-eclampsia (P<0.001). Platelet distribution width (rho = 0.731, p<0.001) and mean platelet

**Funding:** This study was sponsored by the Jimma University (JU), as one of the 1st generations higher Institution University in Jimma, Oromia, Ethiopia financed this study as part of a community research project for research and community service. The funder had no role in the design of the study, data collection, and analysis, interpretation of the data, and preparation of the manuscript.

**Competing interests:** The disclosure of affiliations and interests allows for a more comprehensive and transparent mechanism, resulting in a more accurate and objective appraisal of the work. There should be no sense of struggle for any writer.

volume (rho = 0.674, p<0.001) had statistically significant positive relationships with mean arterial pressure. The best metric for predicting preeclampsia was platelet distribution width (AUC = 0.986; 95%CI; 0.970, 1).

## Conclusions

Platelet indices, including platelet count, mean platelet volume, platelet distribution width, and Platelet crit, have been identified as promising candidate markers for predicting pre-eclampsia in pregnant women. In the future, a serial examination of these indicators during several trimesters of pregnancy should be conducted.

## Introduction

Normal pregnancy's inability to adjust metabolic and hemodynamic changes in a woman's physiology for fetal growth can lead to pregnancy-related problems [1]. Pregnancy-induced hypertension is the most serious of these problems [2]. After hemorrhagic diseases, it is the second-largest cause of maternal death globally [3]. Preeclampsia manifests as new-onset hypertension and proteinuria after 20 weeks of pregnancy [4]. It complicated 5–8% of pregnancies [5], and 9%–26% of global maternal death, a considerable proportion of preterm birth, and maternal and newborn morbidity [6]. It represents 2.32% of all deliveries in Africa [7]. It has a prevalence of 5.47% in Ethiopia [8]. In Ethiopia, it was responsible for 1% of all deliveries, 5% of all women with difficulties, nearly 16% of direct maternal death, and a case fatality rate of 3.6% [9].

From year to year, publications in Ethiopia have identified a rising tendency in maternal problems [10]. To build a better tracking system for antenatal (ANC) programs and early prevention and intervention, preeclampsia must be predicted using reliable indicators [11]. Simple markers can be used as a prognostic tool in management without incurring additional costs [12].

Despite attempts, the specific pathogenic element of preeclampsia is yet unknown [13]. However, the current understanding of preeclampsia is that disruption causes in placentation by various genetic and epigenetic variables [14]. In preeclampsia, endothelial dysfunction can lead to hemostatic changes [15]. To yet, no well-established and foolproof techniques for preventing preeclampsia have been developed [16,17].

The World Health Organization (WHO) defines direction as retrieving current research-based knowledge to provide clinicians with extra pre-eclampsia information [18]. Despite numerous studies over the last decade to create a viable test, and various biochemical markers recently described for predicting preeclampsia, their utility in resource-limited hospitals is questionable, and some are still under investigation [19]. Evaluation of platelet (PLT) indices can be a potential candidate marker in this regard because it is a simple and routinely performed procedure with reduced costs and more accessibility in the clinical laboratory [20].

Evidence links PLT activation with preeclampsia [21], although the trend of employing them in diagnosis is not yet fully developed in our setting. Despite the clinical benefits, research in Ethiopia found that only around 10% of doctors employed the Mean Platelet Volume (MPV) and Platelet Distribution Width (PDW) in medical practice [22].

Furthermore, reports of a large difference in PLT indices between preeclampsia and normal pregnancy have sparked debate, and more study is needed to get a clear picture. In Ethiopia,

there is limited literature comparing PLT indices in preeclampsia and normal pregnancy, and no study has been conducted in the study area, necessitating the current investigation.

## 2. Methods and materials

### 2.1.Study setting, design, and period

An institution-based case-control study was conducted among pregnant women attending Wachemo University Nigist Eleni Mohammed Memorial comprehensive and specialized Hospital, Hosanna southern Ethiopia between the days of January to April 2019. The Zone is located 232 km southwest of Addis Ababa, and 194 km west of Hawassa regional capital city. The hospital was established in 1984 which provides its referral and non-referral services for around 3.2 million populations in its catchment areas. On average, a total of 1979 pregnant women are visiting the hospital annually as per the 2018 Hospital report [23].

### 2.2.Sample size determination

The sample size for this study was calculated using the G-power statistical software version 3.1. The level of significance was set to 0.05. Power of the test: 100(1-β) %, is 90%, which was equal to 0.9 and the effect size of 0.5 The mean PLT count considered for sample size calculation for normal pregnancy was 194.05±45.59 and non-severe pre-eclampsia women were 171.06±36.91 [31]. The sample size obtained was 156. By adding a 15% non-response rate, the final sample size was 180. To increase the accuracy, we used 1:2 allocation ratios; the number of control was twice the number of cases. Therefore, 60 pregnant women with pre-eclampsia (30 non-severe pre-eclampsia and 30 severe pre-eclampsia pregnant women) and 120 with healthy pregnancy as a control total 180 study subjects were recruited.

### 2.3.Sampling procedure

All pregnant women who visit Wachemo University Nigist Eleni Mohammed Memorial comprehensive and specialized Hospital were the source population. An antenatal record was observed if the pregnant woman met the inclusion requirements and gave her consent. After being evaluated by the attending physician, the status was determined(preeclamptic or normotensive) using a code. A blood sample was taken and labeled using the given code. A Mindray BC3000-plus (Mindray®, Shenzen, China) automated hematological analyzer was used to examine the blood sample. Under a microscope, the sample with the low PLT count was inspected. The value of PLT, MPV, PDW, and Platelatecrit(PCT) was recorded in a logbook made specifically for this purpose. Then, during the study period, all preeclampsia and normal pregnant mothers who attended ANC follow-up and were admitted to NEMMCSH were sequentially enrolled. Finally, the data were entered, processed, and the outcome was deduced. Pre-eclamptic pregnant women and healthy normotensive pregnant women, who were volunteers to participate and at ≥20-week gestation, were included in the study, but the pregnant women with Poor past obstetric history (recurrent miscarriage, pre-term labor, intrauterine growth restriction), Gestational or insulin-dependent diabetes and known previous hypertension, history of preeclampsia, renal or hepatic dysfunction, disseminated intravascular coagulation, symptomatic infectious disease, autoimmune conditions such as lupus, took drugs which alter PLT count such as heparin, corticosteroid, interviewing study participants to manage missing information, communicating with attending physician (for those clinically diagnosed cases) were excluded.

## 2.4.Measurement of variables

Platelet indices; platelet count (PLT), Platelet distribution width (PDW), Platelet crit (PCT) and Mean platelet volume (MPV) were dependent variables. Pre-eclampsia, Gestational age, Residence, Maternal age, Body mass index (BMI), Number of delivery and Number of pregnancies were they are predictors variables.

Those volunteer pregnant women of ≥20 weeks of gestation after getting informed consent were clinically examined by the physicians.

**Preeclampsia.**   Defined according to the International Society for the study of hypertension in pregnancy as an increase in blood pressure to at least 140/90 after 20th week of gestation in a previously normotensive woman, combined with protein urea (protein excretion at least 0.3g per 24 hours,+2 protein by dipstick [24].

**Non-Severe preeclampsia.**   Two readings of systolic blood pressure >140–160 & diastolic blood pressure 90-110mmHg 4–6 hours apart, after 20 weeks of gestation and with proteinuria of >300mg/l in 24 hours or up to 2+and with/without edema [24].

Severe Preeclampsia: SBP≥160 & DBP≥110mmHg after 20 weeks of gestation. There may be a severe headache, blurred vision, epigastric pain or right upper quadrant pain, oliguria, Eclampsia, HELLP (Haemolysis, Elevated Liver enzymes, and Low Platelet count) syndrome, elevated serum creatinine, IUGR (Intra Uterine Growth Restriction), and pulmonary edema [24].

**Platelet indices.**   Platelet indices include PLT, MPV, PDW, and PCT between preeclamptic and normotensive pregnant women as well as with the severity of preeclampsia [25].

The Blood Pressure (BP) was measured and recorded using a mercury sphygmomanometer according to the recommendation of Guideline for the management of Hypertensive disorders [7] and as a marker of preeclampsia severity; Mean Arterial Pressure (MAP) was calculated as Diastolic Blood Pressure (DBP) plus 1/3 of the difference between Systolic Blood Pressure (SBP) and DBP [8]. Then after those pregnant women with a SBP of ≥ 140 mmHg or DBP of ≥ 90 mmHg recorded twice 4 hours apart or a single measurement of ≥ 160/110 mmHg, accompanied by significant proteinuria was considered as preeclamptic(cases) and out of this pregnant women, those with the blood pressure of ≥ 160 mmHg (SBP) or 110 mmHg (DBP) and associated proteinuria of ≥ 0.3 grams(+1 on dipstick) and with severity signs in a clinical examination such as new-onset cerebral or visual disturbance, epigastric or right upper quadrant pain and pulmonary edema considered as severe preeclamptic and those whose blood pressure less than 160 mmHg (SBP) or 110 mmHg (DBP) with proteinuria greater than ≥ 0.3, grams(+1 on dipstick), was considered as non-severe preeclampsia but those pregnant women without this feature of hypertension and proteinuria were considered as normotensive(controls). Then, samples of both preeclamptic and normotensive pregnant women that fulfill the inclusion criteria were recruited consecutively. The socio-demographic data such as maternal age, gestational age, gravidity, parity, the residence was collected using a structured questionnaire from both pregnant women with pre-eclampsia (cases) and normal pregnant women (control), and BMI was computed as the ratio of maternal weight in kilograms and the square of maternal height in meters the result can be validated using four ranges: < 20 kg/m2 (underweight), 20 to 26 kg/m2 (normal weight), 26 to 29.9 kg/m2 (overweight) and >30 kg/m2 (obese) for BMI recommended by BilanoVL [9].

The three (3 ml) of venous blood was collected once from both preeclamptic and normal pregnant women by clean venipuncture, using vacutainer tube method, into the commercially prepared concentration of Ethylene Di Amine Tetra Acetic Acid (EDTA) containers following Standard Operating Procedures(SOP). A blood sample was gently mixed to prevent clump and clot formation for PLT indices. Samples were measured in MINDRAY-BC-300Plus,

Shenzhen China within 1 hour of blood collection to determine the value of the PLT indices. The MINDRAY-BC-300Plus performs speedy and accurate analysis of 19 parameters using impedance principle for counting and cyanide-free testing for hemoglobin. After the analysis, the results obtained were print out and registered on registration books. The sample of thrombocytopenia (PLT<150, 0000/µl) was rechecked by examining Wright's stained blood film on a microscope (normal PLT 150,000/µL-450,000/µL) to exclude the error of machine and to evaluate morphological change in PLT. The result with abnormal PLT was reported to physicians before analysis of data to manage the patient [26].

## 2.5.Data quality assurance

The Personal protective equipment was used appropriately while performing the procedure as well as the SOP was followed while collecting the sample and the blood sample was collected by a trained phlebotomist, checked for criteria's like; hemolysis, clotting, volume and collection time, and labeling after collection as well as homogenized by inverting 5–6 times before analysis according to the recommendation of guidelines on the laboratory aspects of assays used in hemostatic and thrombosis [27].

The quality of the sample and reagents was assured based on SOP and the performance of the hematological analyzer was maintained by running three levels of hematology cell controls (Normal, Low, and High) based on the protocol of the laboratory. All the result of PLT below and above reference limit was rechecked by examining Wright stained blood film. The completeness of each data was checked daily. The result of PLT indices was printed and registered on request prepared for this purpose.

## 2.6.Data processing and analysis

The socio-demographic and laboratory data were entered in EPI data version 3, and then the data were exported and analyzed using SPSS version 21. The Kolmogorov-Smirnov normality test was run for checking the distribution of PLT indices. Kruskal-Wallis H test in conjunction with the Mann-Whitney U test was used for comparison of non-normally distributed parameters and the results were presented as median and minimum and maximum values. Before, a post hoc test was done for comparison of PLT indices across the three groups of women (severe preeclampsia, non-severe preeclampsia, and normal pregnant women). Receiver operating curve (ROC) curve was done to determine sensitivity; specificity; Area under the Curve (AUC), and cut-off value for a given PLT indices (PLT, MPV, PDW, and PCT) in discriminating the presence or absence of preeclampsia. Based on estimations of sensitivity and specificity for the cut-off value of the PLT indices, Positive Predictive Value (PPV) and Negative Predictive Value (NPV) for each of them were determined using Medical Calculator (MedCalc©) statistical software version19.0.4, where sensitivity and specificity were obtained from ROC curve and known case with the disease was 60 and known case without disease was 120. Spearman rank-order test was used to evaluate the Correlation between PLT indices with Mean Arterial Pressure (MAP). A P-value of <0.05 was considered as statistically significant.

## 2.7.Ethical approval

The ethical approval was obtained from the Ethical Review Committee (ERC) of the JU Institute of Health Sciences. Before beginning the investigation, permission was obtained from Hadiya Zone Health Office and WCUNEMMCSH hospital. Participants were given written consent about the study, its goals, effects, and the significance of the data before they were enrolled. To ensure confidentiality, all information was rendered anonymous.

## 3. Results

### 3.1.Characteristics of respondents

A total of 180 study participants from two groups were recruited in the study. The first group included 120 normotensive pregnant women and the second group 60 pregnant women with preeclampsia. Out of 60 preeclampsia cases, 30 of them had non-severely pre-eclamptic and the remaining 30 cases were severely preeclamptic. The median (minimum-maximum) ages of the normotensive, non-severe and severe preeclampsia group in full-year was 25.00(20–36), 28.00(18–37), and 28.50(18–39) respectively. In this study, no statistically significant differences were observed between the three groups in age, residence, number of pregnancies (gravidity), number of deliveries (parity), gestational age, and BMI, but there was a significant difference between the three studied groups with regards to SBP, DBP and MAP which increased with severity of preeclampsia ($P<0.001$) (Table 1).

Where, (0 indicate negative for proteinuria, +2 and +3 for results of values of urine dipstick protein. P-value is significant at the level of <0.05, "The result is expressed with median (minimum-maximum) and number (%)".

### 3.2.Platelet indices among pregnant mothers

**3.2.1. Level of thrombocytopenia among study participants.** According to this study, twenty-eight (28) of the study participants were thrombocytopenic (PLT<150×10$^3$/μl) which accounts for 15.6%; whereas, 152 of the total study participants appeared with normal PLT count and accounts for 84.4%. The level of thrombocytopenia in severe preeclampsia cases, cases with non-severe features, and normotensive pregnant women accounts for 6/120, 8/30, and 14/30 for normotensive, non-severe preeclampsia, and severe preeclampsia respectively Table 1 (Fig 1).

**3.2.2. Comparison of platelet indices across preeclampsia and normotensive pregnant women.** The median (min-max) values of PLT and PCT were significantly lower in preeclamptic pregnant women than normotensive women. The value of PLT accounts 170(97–352) ×10$^3$/μl for preeclamptic women and 251(139–445) ×10$^3$/μl for normotensive pregnant women (p<0.001). The value of PCT for the two groups of pregnant women was 0.1530 (0.016–0.292) % for preeclamptic women and 0.1975(0.098–0.398) % for normotensive pregnant women (p<0.001); Whereas MPV and PDW were significantly higher in the

**Table 1. Socio-demographic and clinical characteristics of study participants in NEMMCSH from January 1 to April 3, 2019.**

| Characteristics | | Normotensive (n = 120) | Non-Sever Preeclamptic(N = 30) | Severe-preeclamptic(n = 30) | p-value |
|---|---|---|---|---|---|
| Age in years | | 25(20–36) | 28(18–37) | 29(18–39) | 0.088 |
| Residences | Urban | 64(69.5%) | 16 (17.3%) | 12(13%) | 0.413 |
| | Rural | 56(63.6%) | 14(15.9%) | 18(20.4%) | |
| Gravidity | | 1(1–5) | 2(1–6) | 2(1–6) | 0.896 |
| Parity | | 1(0–4) | 1(0–5) | 0(0–5) | 0.690 |
| BMI(Kg/M$^2$) | | 24.60(21–31) | 25.9(20.2–30.7) | 25.1(20.6–30.8) | 0.84 |
| SBP(Mm/Hg) | | 114(93–137) | 143(130–159) | 160.50(160–170) | <0.001 |
| DBP(Mm/Hg) | | 72(50–93) | 100 (90–119) | 111.50(100–129) | <0.001 |
| MAP(Mm/Hg) | | 85.16(53–119) | 113.83(103.3–131) | 128.00(120.33–141.33) | <0.001 |
| Proteinuria | | 0 | +2(1–3) | +3(1–3) | <0.001 |

Where: BMI (Kg/M$^2$), Body Mass Index in kilogram per meter squares, SBP (Mm/Hg), Systolic Blood Pressure in millimetres per mercury, DBP (Mm/Hg), Diastolic Blood Pressure in millimetres per mercury, and MAP (Mm/Hg), Mean Arterial Pressure in millimetres per mercury.

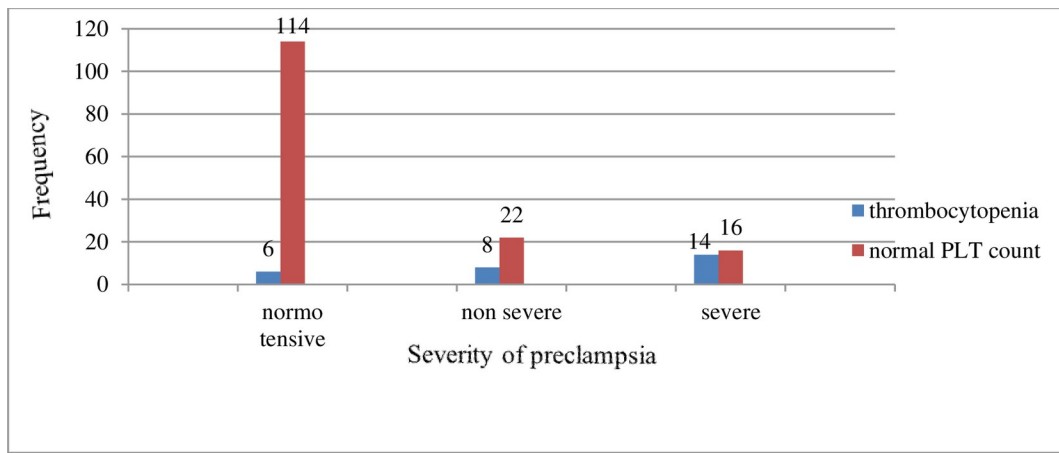

**Fig 1. Frequency of thrombocytopenia among study participants in NEMMCSH from January 1 to April 3, 2019.**

preeclampsia group than the control group. The value of MPV among preeclamptic women was 9.25 (8–12.5) and its value among normotensive pregnant women was 8(6.9–9.3) fl. The level of PDW among preeclamptic women was 16.250(15.5–18) and for normotensive pregnant women, its value was 15(14–16.1) fl with ($p < 0.001$) in the Mann-Whitney U test (Table 2).

**3.2.3. Comparing platelet indices between normotensive, non-severe, and severe pre-eclamptic pregnant women.** In the Kruskal-Wallis H test, MPV and PDW have shown significant differences among the three groups. The values were significantly elevated as the disease severity advances ($p < 0.001$). In this study, the PLT count was significantly decreased as the disease progressed from normal, non-sever to the severe stage with the values 251(139–445), 196.50(110–352), and 155(97–230) for normotensive, non-severe and severe preeclamptic pregnant women, respectively ($p < 0.001$) (Table 3).

**3.2.4. Intergroup analysis of platelet indices across normotensive, non-severe, and severe pre-eclamptic pregnant women.** In Bonferroni pairwise comparison tests between groups, there were statistically significant differences among the normotensive and non-severe preeclampsia group, normotensive and severe preeclampsia groups as well as non-severe preeclampsia group and severe concerning PLT and PCT which showed significantly declining value with the severity of the disease, In contrast, MPV and PDW significantly increased with the severity of preeclampsia (Table 4).

**3.2.5. Co-relational analysis of platelet indices with mean arterial pressure in study participants.** In this study, the spearman rank-order correlation of PLT indices with the MAP was computed to evaluate their association with the severity of the disease. In correlation

**Table 2. Comparisons of platelet indices between normotensive and pre-eclamptic pregnant women in NEMMCSH January 1 to April 3, 2019.**

| Platelet indices | Normotensive pregnant women | Preeclamptic pregnant women | P-value |
|---|---|---|---|
| PLT×10³/μl | 251(139–445) | 170 (97–352) | <0.001 |
| MPV(fl) | 8(6–9.3) | 9.25(8–12.5) | <0.001 |
| PDW(fl) | 15(14–16.1) | 16.2(15–18) | <0.001 |
| PCT (%) | 0.1975(0.098–0.398) | 0.153(0.016–0.292)) | <0.001 |

Where: PLT×10³/μl, Platelate in per microliter, MPV Mean Platelet Volume in femtoliter (fl), PDW (fl) Platelet Distribution Width in femtoliter and PCT (%) Platelate Crit in percentage.

**Table 3. Comparisons of platelet indices among normotensive, non-severe, and severe pre-eclamptic pregnant mothers in NEMMCSH from January 1 to April 3, 2019.**

| Platelet indices | Normotensive pregnant women | Non-severe pre-eclampsia group | Severe pre-eclampsia group | P-Value |
|---|---|---|---|---|
| PLT×10³/Ml | 251(139–445) | 196.5(110–352) | 155(97–230) | <0.001 |
| MPV(fl) | 8(6–9.3) | 9(8–10.4) | 9.6(8–12.5) | < 0.001 |
| PDW(fl) | 15(14–16.1) | 16.0(15–17.1) | 16.5(15.9–18) | <0.001 |
| PCT (%) | 0.1975(0.098–0.398) | 0.166(0.111–0.292) | 0.146(0.016–0.207) | <0.001 |

Where: PLT×10³/μl, Platelate in per microliter, MPV Mean Platelet Volume in femtoliter (fl), PDW (fl) Platelet Distribution Width in femtoliter and PCT (%) Platelate Crit in percentage.

analysis, a MAP showed statistically significant positive correlations with PDW (rho = 0.731, p <0.001), and MPV (rho = 0.674, p<0.001). Moreover, MAP showed significant negative correlation with PLT (rho = -0.503 and PCT (rho = -0.369, p<0.001) (Table 5).

**3.2.6. The diagnostic role of platelet indices for preeclampsia.** The ROC curve analysis was used to determine the optimal cut-off values of PLT indices to the prediction of pre-eclampsia. The analysis showed that PLT can differentiate normotensive pregnant women from preeclamptic pregnant women at a cut-off value ≤of 224x10³/μl with a sensitivity of 88.3%, the specificity of 64.2%, PPV of 71.1%, and NPV of 84.5% with an AUC of 0.858. Whereas MPV can differentiate normotensive pregnant women from preeclamptic pregnant women at a cut off value ≥of 8.55fl with a sensitivity of 86.6%, the specificity of 89.2%, PPV of 88.9%, NPV of 86.94% while PDW can differentiate normotensive pregnant women from pre-eclamptic pregnant women at a cut off value of ≥15.45 with a sensitivity of 98.3%, the specificity of 91.7%, PPV of 67.42%, and NPV of 96.86%. The PCT at a cut-off value of ≤0.1915% with a sensitivity of 83.3%, specificity 52.5%, PPV of 63.68%, and NPV of 75.86% can differentiate normotensive pregnant women from preeclamptic women. PDW has the largest area under the curve (AUC = 0.986; 95%CI (0.970, 1), indicating as it is the best parameter for predicting preeclampsia. The second most important predictor identified was MPV (AUC = 0.954; 95% CI (0.925, 0.984) followed by PLT (AUC = 0.858; 95%CI (0.801, 0.916) (Fig 2 and Table 6).

## 4. Discussion

According to this study, the PLT and PCT were decreased with the severity of pre-eclampsia. Twenty-eight (28) of the study participants were thrombocytopenic which accounts for 15.6%.

**Table 4. Pairwise comparisons of platelet indices among normotensive, non-severe, and severe pre-eclamptic pregnant women in NEMMCSH from January 1 to April 3, 2019(post hock).**

| Platelet Indices | Normotensive | Non-Severe | Severe | *P-Value | **P-Value | ***P-Value |
|---|---|---|---|---|---|---|
| PLT×10³/Ml | 251(139–445) | 196.5(110–352) | 155(97–230) | 0.019 | <0.001 | <0.001 |
| MPV(fl) | 8(6–9.3) | 9 (8–10.4) | 9.6(8–12.5) | < 0.001 | < 0.001 | < 0.001 |
| PDW(fl) | 15(14–16.1) | 16.0((15–17.1) | 16.5(15.9–18) | 0.018 | < 0.001 | < 0.001 |
| PCT (%) | 0.1975(0.098–0.398) | 0.166(0.111–0.292) | 0.146(0.016–0.207) | 0.045 | < 0.001 | 0.004 |
| PCT (%) | 0.1975(0.098–0.398) | 0.166(0.111–0.292) | 0.146(0.016–0.207) | 0.045 | < 0.001 | 0.004 |

**Where**

*P- comparison of non-severe and severe preeclampsia, the p-value was significant at p-value < 0.05.

**p- comparison of severe preeclampsia group and normotensive pregnancy.

***p- comparison of non-severe preeclampsia group and normotensive pregnancy.

Where: PLT×10³/μl, Platelate in per microliter, MPV Mean Platelet Volume in femtoliter (fl), PDW (fl) Platelet Distribution Width in femtoliter and PCT (%) Platelate Crit in percentage.

**Table 5. Correlation between Platelets indices with Mean Arterial Pressure for the study participants in Wachemo University Nigist Eleni Mohammed Memorial comprehensive and specialized Hospital from January 1 to April 3, 2019.**

| | Platelet Indices | | | |
|---|---|---|---|---|
| | **PLT** | **MPV** | **PDW** | **PCT** |
| **MAP(mm/Hg)** | rho -0.503 | 0.674 | 0.731 | -0.369 |
| | <0.001 | <0.001 | <0.001 | <0.001 |

Correlation is significant at the 0.01 level (2-tailed), where: MAP (Mm/Hg), Mean Arterial Pressure in millimetres per mercury.

The values of MPV and PDW were significantly elevated with preeclampsia severity. The MAP showed statistically significant positive correlations with PDW and MPV, in contrast, it showed a significant negative correlation with PLT and PCT. At the cut-off value of $\leq$ 224x10$^{3}$/µl with a sensitivity of 88.3%, the specificity of 64.2%, PPV of 71.1%, and NPV of 84.5%, PLT differentiate pre-eclamptic women from normotensive with an AUC of 0.858. Whereas the cut-off value $\geq$of 8.55fl with a sensitivity of 86.6%, the specificity of 89.2%, PPV of 88.9%, NPV of 86.94% accounted for MPV. The PDW can differentiate normotensive pregnant women from pre-eclamptic women at a cut-off value of $\geq$15.45fl with a sensitivity of 98.3%, the specificity of 91.7%, PPV of 67.42%, and NPV of 96.86%. The PCT at a cut-off value of $\leq$0.1915% with a sensitivity of 83.3%, specificity 52.5%, PPV of 63.68%, and NPV of 75.86% can differentiate normotensive pregnant women from pre-eclamptic women.

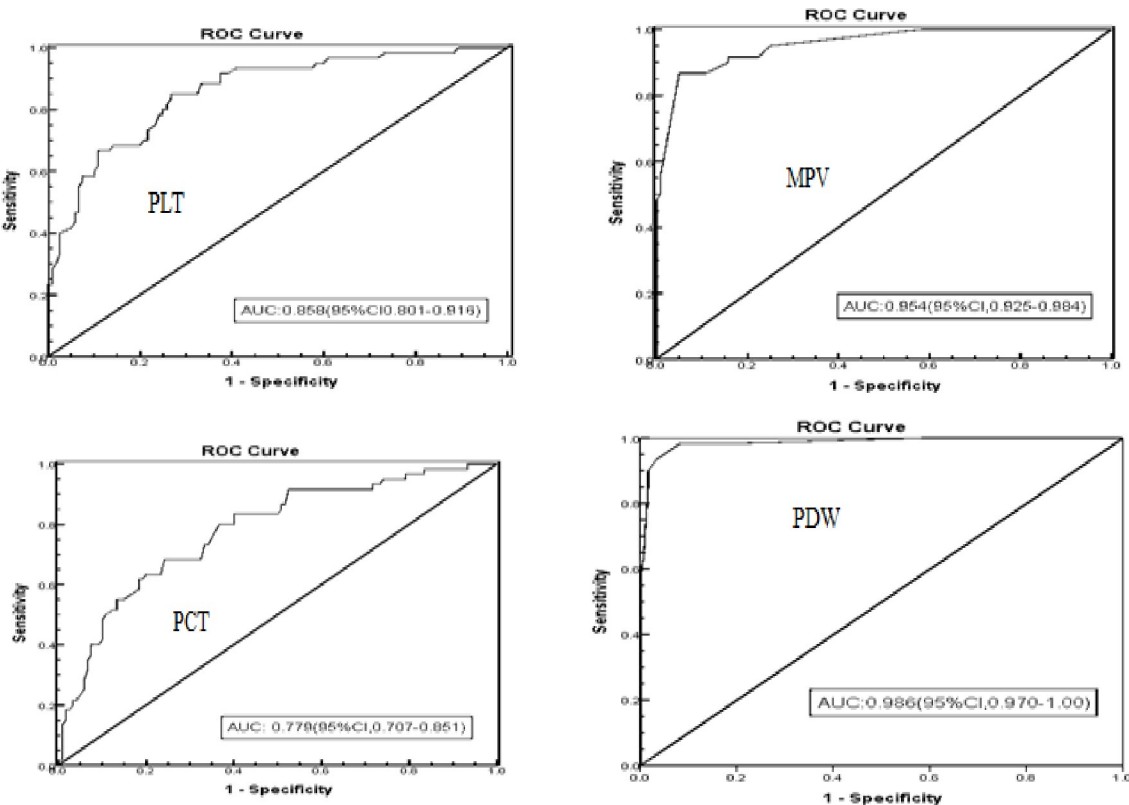

**Fig 2. The ROC curve analysis of PLT indices for study participants in NEMMCSH from January 1 to April 3, 2019.**

**Table 6. The Diagnostic values of platelet indices for preeclampsia among study participants in Wachemo University Nigest Eleni Mohammed Memorial Referral Hospital from January 1 to April 3, 2019.**

| Platelet indices | Sensitivity(% | Specificity (%) | PPV (%) | NPV (%) | Cut-off value | AUC | (95%CI) | P-Value |
|---|---|---|---|---|---|---|---|---|
| PLT× $10^3$/Ml | 88.3 | 64.2 | 71.1 | 84.5 | ≤224 | 0.858 | (0.801,0.916) | <0.001 |
| MPV(fl) | 86.6 | 89.2 | 88.9 | 86.94 | ≥8.55 | 0.954 | (0.925,0.984) | <0.001 |
| PDW(fl) | 98.3 | 91.7 | 67.42 | 96.86 | ≥15.45 | 0.986 | (0.984, 1.000) | <0.001 |
| PCT (%) | 83.3 | 52.5 | 63.68 | 75.86 | ≤0.1915 | 0.779 | (0.707,0.851) | <0.001 |

P-value is significant at level <0.05. Where: PLT×$10^3$/µl, Platelate in per microliter, MPV Mean Platelet Volume in femtoliter (fl), PDW (fl) Platelet Distribution Width in femtoliter and PCT (%) Platelate Crit in percentage.

According to a current study, PLT and PCT were found to be significantly lower in the pre-eclampsia group, while MPV and PDW were found to be significantly greater. Other researchers [21,28–32] reported the same findings. Increased vascular reactivity and PLT activation following increased strain on placental endothelium due to preeclampsia, which results in PLT consumption [15], could be the underlying cause for PLT indices' variance between pre-eclamptic and pre-eclamptic non-preeclamptic women. In contrast to our findings, research conducted in Turkey found no significant change in PLT between severe preeclampsia and normotensive pregnant women [33]. Another study [20,34,35] backed up this conclusion. The discrepancies could be due to the maximum time between sample collection and analysis, which causes time-dependent platelet activation, the effect of anticoagulant employed variation in the hematological analyzer, the existence of confounders, and the limited sample size.

In our study, the PLT cutoff value was 224×$10^3$/µl to distinguish preeclamptic women from normotensive pregnant women, with sensitivity (SE), specificity (sp),), positive predictive value (PPV), and negative predictive value (NPV) of 88.3%, 64.2%, 71.15%t, and 84.5%, respectively, and an AUC of 0.858. The findings resemble those of studies conducted in Brazil and Egypt [31,36]. A slightly higher cut-off value of 248×$10^3$/µl was noted in a study conducted in Saudi Arabia [37]; however, a study conducted in Egypt and Turkey [30,33] revealed a lower cut-off value. The discrepancy in the hematological analyzer could be the cause and certain analyzers tend to exaggerate some metrics while underestimating others [38].

There were 28 cases of thrombocytopenia in the current study, with 6 cases in the normotensive group, 8 cases in the non-severe group, and 14 cases in the severe pre-eclampsia group. Similarly, another study [29] reported a similar finding of 30 cases of thrombocytopenia; among these, 9 cases were in non-severe pre-eclampsia and 21 in severe pre-eclampsia. On the contrary, other studies [39,40] showed a slightly higher number of cases, 33 cases (56%) within the severe pre-eclampsia group. Other studies [41,42] found 11 and 16 cases of severe pre-eclampsia, respectively. In contrast to our findings; other researchers [36,37] found that all study participants had a normal range of PLT. However, our research found that 5% of normal pregnant women with normal blood pressure have thrombocytopenia. Our findings could be attributed to gestational thrombocytopenia caused by hemodilution, aggregation, and PLT consumption, although they are more closely linked to the severity of hypertension in pre-eclampsia.

Pre-eclamptic pregnant women had a lower PCT value than normotensive pregnant women, according to the current study. This is in line with findings from research conducted in Korea [21], and Brazil [36]. In contrast to the current study, studies conducted in Turkey [20] and Sudan [34] found no significant difference between pre-eclamptic and normotensive pregnant women regarding PCT. The discrepancies could be attributable to variances in the hematological analyzer, sample processing delays, or the anticoagulant utilized. Because time-

dependent activation and EDTA-induced PLT activation may increase the value of MPV, which in turn may affect the PCT calculation.

According to our findings, PCT may distinguish normotensive pregnant women from pre-eclamptic pregnant women, at a cut-off value of 0.1915% with a sensitivity of 68.3% and specificity of 69.2% has an AUC of 0.776 (p 0.001). As a result, it's a good predictor of pre-eclampsia. Studies were done in Brazil [36] and Korea [21] both reported a similar cut-off value. As a result of our findings, PCT appears to decline with the severity of pre-eclampsia. Because a PCT rate of less than 0.1% is an indirect indicator of PLT transfusion and a more specific determinant in thrombocytopenic patients, including pre-eclampsia [43], it is a more specific determinant than the PLT number.

The MPV was shown to be higher in the current study's findings. According to the current investigation, MPV levels were greater in pre-eclamptic women than in normotensive women. Our findings are consistent with those studies conducted in Turkey [20], Korea [21], and India [44]. This could be related to young PLT producing significant amounts of marrow as a compensation mechanism for PLT consumption and degradation in pre-eclampsia [37]. However, other investigations [35,37,45,46] have established the inconsistency of the results. This might be b PLT-generated metrics are highly reliant on the specific technology, modified by the anticoagulant, and take a long time to analyze. In impedance counting, the MPV rises over time as the PLT swells in EDTA, and can climb by 7.9% in 30 minutes [38].

From ROC analysis, MPV can distinguish pre-eclamptic pregnant women from normotensive pregnant women with a sensitivity of 86.6%, specificity of 89.2%, PPV of 88.9%, and NPV of 8.9% at the cut-off value of 8.55fl. This value has an AUC of 0.954, according to the study. A study conducted in Turkey [22] came up with a similar conclusion. Studies in Egypt [30,31] and Brazil [36] reported a slightly higher figure. As a result of this discovery, MPV may be a good potential marker to distinguish the presence or absence of pre-eclampsia in the literature [47].

The current investigation found that pre-eclamptic pregnant women had a much higher PDW value than normotensive pregnant women, with a progressive increase as disease severity rose. Similar findings have been reported from Brazil [28] and India [44], as well as Korea [21]. However, a similar study in Turkey and Saudi Arabia [33,37] found no significant difference between pre-eclamptic and normotensive women, contrary to our findings. A conflicting outcome could be attributed to time delay-induced PLT activation, leading to an increase in PDW levels. Other research found similar results to ours [30–32,34,45,48–50].

According to the current investigation findings, the cut-off value of 15.4fl for PDW has a sensitivity of 98.35%, specificity of 91.7%, PPV of 67.42%t, and NPV of 96.86% (p<0.001). The AUC for this value is 0.986. A study conducted in Korea [21], Brazil [28], and Egypt [31] found a similar cut-off value, which is consistent with our findings. However, a larger cut-off value was observed from Turkey [20], contrary to our findings. Our investigation found a little higher cut-off value than an Egyptian study, which found a cut-off value of 12.6fl with the sensitivity of 90%, specificity of 92%, PPV of 91.8%, NPV of 90.2%, and AUC of 0.886 to distinguish normotensive women from non-severe-preeclamptic women [30]. As a result, according to the general rule of thumb for interpreting AUC, for evaluating the diagnostic ability of a test in discriminating the true disease status of a patient, recommended by Yang S, Berdine G [50], PDW and MPV were the outstanding parameters for the prediction of pre-eclampsia due to their large AUC.

PDW and MPV were found to have a strong positive connection with MAP in our study. However, there was a strong negative connection between MAP and PLT. A study conducted in Asia [49] came to the same conclusion. A study conducted in Korea found a statistically significant positive association between PDW and MAP, while other PLT indices did not

demonstrate a significant correlation, contrary to our findings [29]. However, another investigation identified MPV as a pre-eclampsia severity marker [51]. Another study conducted in Egypt [31] and Gondar [32] came to similar conclusions as ours.

As a result, rather than depending solely on PLT count, combining all of the PLT indices in combination to diagnose pre-eclampsia may provide a more valid diagnosis since they compensate for each other's limitations.

## 5. Conclusion

PLT parameters such as PLT, MPV, PDW, and PCT have been found as potential candidate markers for pre-eclampsia prediction. They may serve as diagnostic criteria for pre-eclampsia. The severity of pre-eclampsia was associated with increased MPV and PDW and a decrease in PLT and PCT. The PDW with the highest AUC was the most important measure in predicting pre-eclampsia, followed by the MPV. The PLT indices such as PLT, MPV, PDW, and PCT should be part of the routine antenatal investigation.

Large scale longitudinal study should be conducted in the study area for serial analysis of PLT indices throughout different trimesters to evaluate whether it is possible to predict pre-eclampsia early.

## Acknowledgments

Local health managers, data collectors, supervisors, and Jimma University (JU) deserve special thanks, which are gratefully acknowledged.

## Author Contributions

**Conceptualization:** Solomon Gebre Bawore, Wondimagegn Adissu, Berhanu Niguse, Yilma Markos Larebo, Nigussie Abebe Ermolo, Lealem Gedefaw.

**Data curation:** Solomon Gebre Bawore, Wondimagegn Adissu, Berhanu Niguse, Yilma Markos Larebo, Nigussie Abebe Ermolo, Lealem Gedefaw.

**Formal analysis:** Solomon Gebre Bawore, Wondimagegn Adissu, Berhanu Niguse, Yilma Markos Larebo, Nigussie Abebe Ermolo, Lealem Gedefaw.

**Funding acquisition:** Solomon Gebre Bawore.

**Investigation:** Solomon Gebre Bawore, Wondimagegn Adissu, Berhanu Niguse, Yilma Markos Larebo, Nigussie Abebe Ermolo, Lealem Gedefaw.

**Methodology:** Solomon Gebre Bawore, Wondimagegn Adissu, Berhanu Niguse, Yilma Markos Larebo, Nigussie Abebe Ermolo, Lealem Gedefaw.

**Project administration:** Solomon Gebre Bawore, Wondimagegn Adissu, Berhanu Niguse, Yilma Markos Larebo, Nigussie Abebe Ermolo, Lealem Gedefaw.

**Resources:** Solomon Gebre Bawore, Wondimagegn Adissu, Berhanu Niguse, Yilma Markos Larebo, Nigussie Abebe Ermolo, Lealem Gedefaw.

**Software:** Solomon Gebre Bawore, Wondimagegn Adissu, Berhanu Niguse, Yilma Markos Larebo, Nigussie Abebe Ermolo, Lealem Gedefaw.

**Supervision:** Solomon Gebre Bawore, Wondimagegn Adissu, Berhanu Niguse, Yilma Markos Larebo, Nigussie Abebe Ermolo, Lealem Gedefaw.

**Validation:** Solomon Gebre Bawore, Wondimagegn Adissu, Berhanu Niguse, Yilma Markos Larebo, Nigussie Abebe Ermolo, Lealem Gedefaw.

**Visualization:** Solomon Gebre Bawore, Wondimagegn Adissu, Berhanu Niguse, Yilma Markos Larebo, Nigussie Abebe Ermolo, Lealem Gedefaw.

**Writing – original draft:** Solomon Gebre Bawore, Wondimagegn Adissu, Berhanu Niguse, Yilma Markos Larebo, Nigussie Abebe Ermolo, Lealem Gedefaw.

**Writing – review & editing:** Solomon Gebre Bawore, Wondimagegn Adissu, Berhanu Niguse, Yilma Markos Larebo, Nigussie Abebe Ermolo, Lealem Gedefaw.

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
