## [Decision Letter · Decision Letter 0]

9 Aug 2021

PONE-D-21-22141

A pattern of Platelet Indices as a Potential Marker for Prediction of Pre-eclampsia among Pregnant Women Attending Wachemo University Nigist Eleni Mohammed Memorial Comprehensive and Specialized Hospital, Southern Ethiopia

PLOS ONE

Dear Dr. Bawore,

Thank you for submitting your manuscript to PLOS ONE. After careful consideration, we feel that it has merit but does not fully meet PLOS ONE’s publication criteria as it currently stands. Therefore, we invite you to submit a revised version of the manuscript that addresses the points raised during the review process.

We look forward to receiving your revised manuscript.

Kind regards,

Linglin Xie

Academic Editor

PLOS ONE

Journal Requirements:

[Local health managers, data collectors, and supervisors deserve special thanks. The study's financing was provided by Jimma University (JU), which is gratefully acknowledged.]

 [The JU, a 1st generation higher institution in Jimma, Oromia, Ethiopia, financed this study as part of a community research project for research and community service. The funder had no involvement in the study's design, data collection, analysis, data interpretation, or manuscript production.]

[The disclosure of affiliations and interests allows for a more comprehensive and transparent mechanism, resulting in a more accurate and objective appraisal of the work. There should be no sense of struggle for any writer.]

7. Please note that in order to use the direct billing option the corresponding author must be affiliated with the chosen institute. Please either amend your manuscript to change the affiliation or corresponding author, or email us at plosone@plos.org with a request to remove this option

8. PLOS requires an ORCID iD for the corresponding author in Editorial Manager on papers submitted after December 6th, 2016. Please ensure that you have an ORCID iD and that it is validated in Editorial Manager. To do this, go to ‘Update my Information’ (in the upper left-hand corner of the main menu), and click on the Fetch/Validate link next to the ORCID field. This will take you to the ORCID site and allow you to create a new iD or authenticate a pre-existing iD in Editorial Manager. Please see the following video for instructions on linking an ORCID iD to your Editorial Manager account: https://www.youtube.com/watch?v=_xcclfuvtxQ

9. Please amend your list of authors on the manuscript to ensure that each author is linked to an affiliation. Authors’ affiliations should reflect the institution where the work was done (if authors moved subsequently, you can also list the new affiliation stating “current affiliation:….” as necessary).

Reviewers' comments:

Reviewer's Responses to Questions

**Comments to the Author**

1. Is the manuscript technically sound, and do the data support the conclusions?

Reviewer #1: Yes

Reviewer #2: Yes

2. Has the statistical analysis been performed appropriately and rigorously? 

Reviewer #1: Yes

Reviewer #2: Yes

3. Have the authors made all data underlying the findings in their manuscript fully available?

Reviewer #1: Yes

Reviewer #2: Yes

4. Is the manuscript presented in an intelligible fashion and written in standard English?

Reviewer #1: Yes

Reviewer #2: No

5. Review Comments to the Author

Reviewer #1: Title: A pattern of Platelet Indices as a Potential Marker for Prediction of Pre-eclampsia among Pregnant Women Attending Wachemo University Nigist Eleni Mohammed Memorial Comprehensive and Specialized Hospital, Southern Ethiopia

Comments:

1) Title : Too long , please delete the setting name and replace it for instance (A pattern of Platelet Indices as a Potential Marker for Prediction of Pre-eclampsia among Pregnant Women Attending A tertiary hospital at Ethiopia !!!!!)

2) Abstract:

1. The practice of platelet indices in the diagnosis of preeclampsia is not yet extensive in Ethiopia and there is little information on platelet patterns in preeclampsia and normal pregnancy!!! : it is not an evidence based method to diagnose preeclampsia using platelet indices for most of obstetric settings and of course including Ethiopia .

Please rephrase this statement

2. The purpose of this study was to assess the pattern of platelet indices for the prediction of preeclampsia in pregnant: Prediction means you have to follow a large number of pregnant women whom you suspect to have PE in the future time, then to assess their platelet indices frequently until you have reached a level that can give you a clue that she has developed PE !!! . This is not what you have done! You have already diagnosed women having PE and their platelet indices was assessed and showed a difference in comparison to the normal group.

This mostly follows the severity of PE not as a diagnosis or prediction for future time .

My suggestion is to change your aim to : The purpose of this study was to assess the pattern of platelet indices in women with preeclampsia in your setting

3) Patients and Methods :

1. You need to define cases of preeclampsia, mild and sever eclampsia in a separate sub section giving the reference at the end. Also definitions for variables used in socieo demographic characters for instance BMI!!!!

2. What was your reference to identify all the platelet indices as normal? Which reference you have used to be abnormal? And what do you meant by platelet indices?

3. How many times you have estimated the platelet indices? I assume once!!!you have to record it in the methods section

4) Result section:

1. It seems that your design of the study is not a cross sectional study!!!! it is a case control study , you have identified your criteria to involve cases and controls before starting .Please change it in the whole text to a case control study .Do not use a comparative !!!

2. Again define how did you categorized your cases to sever and mild or severer and non-sever? adding a reference to them and make it the same words in the text and in the tables

3. Add a footnotes for all the tables including the full name of the abbreviations

Reviewer #2: This manuscript is sound in experimental design and draws reasonable conclusions from the data collected. This work provides novel insight into the mechanisms that may advance detection of preclampsia. However, there are consistent grammatical errors that compromise understanding of the methodology and findings. It is my opinion that this manuscript needs to be copyedited and reviewed to ensure that those edits retain the original meaning before full acceptance.

6. PLOS authors have the option to publish the peer review history of their article (what does this mean?). If published, this will include your full peer review and any attached files.

Reviewer #1: **Yes: **Shahla Alalaf

Reviewer #2: No

---

## [Author Response · Author response to Decision Letter 0]

24 Aug 2021

all things attached on responses to the reviewers and editors

we doesn't provide the correct grant numbers for the research article we study in the ‘Funding Information’ section because the university provide the fund for data collectors and supervisors to facilitate the research work.

---

## [Decision Letter · Decision Letter 1]

21 Oct 2021

A Pattern Of Platelet Indices As A Potential Marker For Prediction Of Pre-Eclampsia Among Pregnant Women Attending A Tertiary Hospital , Ethiopia: A Case-Control Study

PONE-D-21-22141R1

Dear Dr. Bawore,

We’re pleased to inform you that your manuscript has been judged scientifically suitable for publication and will be formally accepted for publication once it meets all outstanding technical requirements.

Kind regards,

Linglin Xie

Academic Editor

PLOS ONE

Additional Editor Comments (optional):

Reviewers' comments:

Reviewer's Responses to Questions

**Comments to the Author**

1. If the authors have adequately addressed your comments raised in a previous round of review and you feel that this manuscript is now acceptable for publication, you may indicate that here to bypass the “Comments to the Author” section, enter your conflict of interest statement in the “Confidential to Editor” section, and submit your "Accept" recommendation.

Reviewer #1: All comments have been addressed

Reviewer #2: All comments have been addressed

2. Is the manuscript technically sound, and do the data support the conclusions?

Reviewer #1: Yes

Reviewer #2: Yes

3. Has the statistical analysis been performed appropriately and rigorously? 

Reviewer #1: Yes

Reviewer #2: Yes

4. Have the authors made all data underlying the findings in their manuscript fully available?

Reviewer #1: Yes

Reviewer #2: Yes

5. Is the manuscript presented in an intelligible fashion and written in standard English?

Reviewer #1: Yes

Reviewer #2: Yes

6. Review Comments to the Author

Reviewer #1: (No Response)

Reviewer #2: (No Response)

7. PLOS authors have the option to publish the peer review history of their article (what does this mean?). If published, this will include your full peer review and any attached files.

Reviewer #1: **Yes: **Shahla Alalaf

Reviewer #2: No

---

## [Editor Report · Acceptance letter]

26 Oct 2021

PONE-D-21-22141R1 

A Pattern Of Platelet Indices As A Potential Marker For Prediction Of Pre-Eclampsia Among Pregnant Women Attending A Tertiary Hospital, Ethiopia: A Case-Control Study 

Dear Dr. Bawore:

I'm pleased to inform you that your manuscript has been deemed suitable for publication in PLOS ONE. Congratulations! Your manuscript is now with our production department. 

Kind regards, 

on behalf of

Dr. Linglin Xie 

Academic Editor

PLOS ONE